# Renal cancer survival in clear cell renal cancer compared to other types of tumor histology: A population-based cohort study

Teesi Sepp[1,2]*, Antti Poyhonen[3], Anneli Uusküla[4], Andres Kotsar[5‡],
Thea Veitonmäki[6‡], Teuvo L. J. Tammela[6,7‡], Aleksei Baburin[4,8‡], Teemu J. Murtola[6,7]

**1** University of Tartu, Faculty of Medicine, Tartu, Estonia, **2** North Estonia Medical Centre, Tallinn, Estonia, **3** Centre for Military Medicine, Finnish Defence Forces, Riihimäki, Finland, **4** Institute of Family Medicine and Public Health, University of Tartu, Tartu, Estonia, **5** Tartu University Hospital, Tartu, Estonia, **6** Tampere University, Faculty of Medicine and Health Technology, Tampere, Finland, **7** TAYS Cancer Center, Department of Urology, Tampere, Finland, **8** National Institute for Health Development, Tallinn, Estonia

☯ These authors contributed equally to this work
‡ These authors also contributed equally to this work.
* teesisepp@gmail.com

## Abstract

### Introduction

Renal cancer (RC) presents a challenge with increasing incidence and mortality. This study compares RC cancer-specific survival (CSS) and overall survival (OS) based on histological type.

### Study design

This population-based retrospective cohort study covers 1995–2017, utilizing cases from the Finnish Cancer Registry. Comorbidity, procedure and treatment information from 1995–2018 were obtained from the national health care registry, while death data originated from the national death certificate registry.
RC cases were categorized by histology to analyze CSS and OS via Fine and Gray's proportional sub-hazards model and Cox regression (adjusting for age, tumor extent, Charlson comorbidity index and treatment).

### Results

The final cohort included 14,413 patients, predominantly ccRCC (75.5%), followed by papillary RCC (pRCC),5.8% and chromophobe RCC (chRCC) 2.1%. Univariate analysis showed better OS for non-ccRCC patients, with 5-year survival (5ySR) of 72.6% (95% CI 70.3–74.7%), compared to ccRCC (62.7%, 95%CI 61.8–63.5). Among non-ccRCC, the 5ySRs were as follows: pRCC 74.3% (95%CI 71.2–77.2), chRCC 82.2% (95% CI 77.2–86.1), sarcomatoid variants (sarcRC) 29.2% (95% CI 20.6–38.3), and

**Data availability statement:** The data used in this study are not publicly available because they include personal-level information, and specific permission from the national authority FINDATA is required to access them. The study protocol was reviewed and approved by the ethics committee of the Pirkanmaa hospital district, as well as by the administrators of each health care registry used in this study. The study utilized solely routinely collected data from national health-care registries, approvals and consents were obtained from registry keepers, Finnish Institute for Health and Welfare (THL) and Social Insurance Institution of Finland (SII).There are legal restrictions on sharing a de-identified data set according to the Finnish Law 488/1999 on Medical Research. Access to data can be applied at https://findata.fi/en/.

**Funding:** The author(s) received no specific funding for this work.

**Competing interests:** The authors have declared that no competing interests exist.

collecting duct carcinoma (CDC) 23.5% (95%CI 7.3–44.9). Non-ccRCC showed improved CSS compared to ccRCC (sHR 0.69, 95% CI 0.60–0.78). Favorable CSS for chRCC (sHR 0.28, 95% CI 0.18–0.43) and pRCC (sHR 0.66, 95% CI 0.56–0.78), while sarcRC (sHR 1.83, 95% CI 1.36–2.46) and CDC (sHR 3.19, 95% CI 2.01–5.08) showed poorer CSS. Overall, non-ccRCC had a better prognosis, driven by pRCC and chRCC, whereas sarcRC and CDC had poor prognoses. CSS has improved over time, with a 62% reduction in death risk since 1995.

## Conclusion

Our results demonstrate that the histological subtype is a powerful predictor of survival. Histology should be used more in clinical decision-making. Having a histological confirmation would tailor the selection of treatment: surgical management for aggressive and ccRCC and conservative management for less aggressive histological types.

## Introduction

Renal cancer (RC) was the eighth most common cancer diagnosed in 2020 [1]. Globocan 2022 reported 434,840 new cases of RC diagnosed yearly and 155,953 deaths related to RC [2]. In Europe, RC mortality increased among males but remained stable among females in the past decade [3]. In Finland the age-standardized incidence of RC diagnosis was 9.4/100, 000 for men and 5.8/100,00 for women in 2003–2007. Age-standardized mortality for the same period was 4.1/100,000 for men and 2.0/100,000 for women [4]. In a more recent 2018 publication, ASR incidence for RC among the Finnish population was 7.9/100,000 [5]. According to the Finnish cancer registry, the renal cancer rate per 100,000 people in 2022 was 18.57 [6]. It has been suggested that the incidence would increase to 14% by 2030 and another 7% by 2040 [7].

Several studies and prediction models have implied that some of the strongest predictors for RC survival are RC size and TNM [8]. Moreover, one population-based study states that after surgery, the prognosis for clear-cell renal cell carcinoma, papillary renal cell carcinoma, and chromophobe renal cell carcinoma does not depend on tumor histology but on the tumor TNM, Fuhrman grade, patient age, and treatment year [9].

In the last three decades, RC diagnosis has been influenced by the growing availability and utilization of medical imaging for various conditions. This has resulted in an increased incidence of renal tumors and extended lead times due to the downgrading of tumor stage at diagnosis [10–12]. Stage downshift has led to 70% of RC being diagnosed at stage I and it is thought that after 2015 stage downshift has stabilized [13].

RC is a complex and diverse disease, with a wide range of histological subtypes and underlying molecular-level mechanisms, which should be back in focus as stage downshift may have ended. RC is traditionally classified into three main groups: clear

cell (ccRCC, accounting for 80% of all RCCs), papillary (pRCC), and chromophobe (chRCC). This categorization does not fully capture the true heterogeneity of RCC [14,15].

In reality, each subtype encompasses a spectrum of histological features and molecular alterations, resulting in a wide range of clinical courses, responses to treatment, and mortality. This aspect of histological heterogeneity is noted as having significance in the metastatic setting [16]; it is not used as a tool before advising local RC treatment.

Histology plays a significant role in RC prognosis. Some rare subtypes, such as collecting duct renal cell carcinoma (CDC) and sarcomatoid variants (sarcRC) (a term used in previous classifications), are known to be associated with more aggressive disease [16–18]. In a US-based study, CDC patients exhibited 40–90% higher cancer-specific mortality (CSM) compared to their counterparts with aggressive ISUP4 (International Society of Urological Pathology grade 4) ccRCC [10,19]. Several studies have confirmed that ccRCC (compared to pRCC or chRCC) is associated with a higher risk for metastasis and CSM [20].

Current treatment guidelines rely primarily on trials with ccRCC cases, as evidence of rare subtypes is scant. As we transition into the era of personalized treatment plans, enhancing our understanding of RCC subtypes is a crucial research objective. Furthermore, it remains unclear whether the stage downshift phenomenon, observed to improve CSS in ccRCC significantly, exerts a similar influence on outcomes in other histological subtypes. We aimed to fill this gap by studying real-life data based on the Finnish population to see the outcomes of current care and survival trends in recent decades.

In this study, we compare overall survival (OS) and cancer-specific survival (CSS) between ccRCC and non-ccRCC in a population-based cohort from Finland. Furthermore, we assess temporal survival trends in recent decades to determine if survival has evolved differently in ccRCC and non-ccRCC.

## Materials and methods

### Study design

A retrospective cohort study linking data from national registries.

### Setting

The data is sourced from nationwide, population-based registries in Finland. A personal unique identification number was used to link the data across the registries. Study period: We included all RC cases from 1995 to 2017 and ended the follow-up on 31 Dec 2018.

### Data sources and definitions

RC cases were identified by the ICD-10 code (C64) from the Finnish Cancer Registry (FCR). Data for the registry is gathered via mandatory reports on all cancer diagnoses made at Finnish healthcare units. FCR collects comprehensive information on the primary cancer site, age (at the time of the cancer diagnosis), gender, histological subtype of diagnosed cancer, laterality, date of diagnosis and data of the chosen treatment method (surgical, conservative) [21]. Data on tumor extent were limited, and we employed the following staging categorization: localized, advanced, or metastatic disease. Data on concomitant comorbidities, diagnostic biopsy and treatment modalities (partial or total nephrectomy and other options for the metastatic setting) were gathered from the national Health Care Registry (HILMO) for the period 1995–2018. HILMO collects and records all diagnoses from in- and outpatient hospital visits and medical procedures performed on patients, using the Nordic Classification of Procedures codes [22]. We used HILMO data to calculate the Charlson comorbidity index for each RC patient and as it has more precise data on chosen treatment methods, it was used to describe the surgical technique (partial or total nephrectomy) [23].

Renal cancer cases were categorized into clear-cell renal cell cancer (ccRCC), papillary renal cell cancer (pRCC), chromophobe renal cell cancer (chRCC), collecting duct carcinoma (CDC) and sarcomatoid renal cancer (sarcRC), and

other known (very rare) subtypes (otherRC). All histopathological diagnoses were made according to the standards of the time period, no reclassifications were done. Since the study period extended over two decades, we have seen many changes in the histopathological classification system. For example, after recent changes in the World Health Organization Classification of Kidney Tumours 2022, sarcomatoid renal cell carcinoma is not a distinct histologic entity. Instead, it describes high-grade transformation in different subtypes of renal cell carcinoma, which is usually observed with ISUP 4 [24]. Due to limitations in the available ISUP grade data, in the present study, cases were classified as sarcomatoid RCC if the term 'sarcomatoid' was explicitly mentioned within the histopathological diagnosis. Cases with unknown histology were excluded from the analysis.

The period of cancer diagnosis was categorized into the following intervals: 1995–1999, 2000–2004, 2005–2009, 2010–2014, and 2015–2017(last 3 years).

The final set of covariates included age at diagnosis, sex, extent of disease at diagnosis, cancer histological subtype, Charlson's score, the period of diagnosis, and treatment received (surgery or else).

The survival interval was defined as the time from the date of renal cancer diagnosis to either the date of death or the date of censoring at the end of the follow-up period (December 31 2018) and presented a 5-year survival rate with 95 percent confidence intervals (CI).

The primary outcome was renal cancer-specific survival, defined as the time to death specifically attributed to renal cancer. CSS was chosen as the primary outcome [25] because it provides a more precise assessment of mortality directly related to the disease, compared to overall survival (OS), which various competing causes of death may influence [26]. The secondary outcome was overall survival, defined as the time to death from any cause. We obtained data on deaths, including the date and cause of death (primary, immediate, and contributory causes, coded as ICD-10 codes) from the national death certificate registry managed by Statistics Finland. Deaths occurring before the end of 2018 were included. RC-specific deaths were defined as those with the ICD-10 code C64 recorded as the primary cause.

## Statistical analysis

We calculated descriptive statistics for all covariates as frequencies for the categorical variables and median with IQR for continuous variables. The demographic characteristics and tumor extent at the diagnosis and treatment of the renal cancer histological subtypes are described, and crude all-cause and renal cancer-specific mortality rates (per 10 0000) are presented. Comparative analyses between ccRCC and non-ccRCC subtypes were performed using the chi-square test for categorical variables and the Mann-Whitney U-test for continuous variables.

Overall survival was defined as the time from diagnosis to death by any cause. Univariable overall survival analysis was performed using the Kaplan-Meier method. Log-rank tests were used to calculate p-values. Multivariable Cox proportional hazards regression models (adjusted for Charlson's comorbidity score (median), age, gender, the extent of disease at diagnosis, modality of treatment and period) were used to assess associations between the histological types of RC and all-cause mortality, calculating hazard ratios (HRs) and their 95% CIs. Multivariable Cox regression analysis was also used to assess the impact of treatment (surgical vs conservative) on CSS by histological subtype in RC. We did not perform imputation for missing data.

Follow-up time was calculated starting from the date of RC diagnosis until the occurrence of death or the common closing date of 31 Dec 2018, whichever occurred first.

We used the cumulative incidence function (CIF) and 5-year survival to describe cause-specific survival and Gray's test to test for cause-specific survival differences. Renal cancer-specific mortality and other cause mortality were considered competing events [27]. We used the cumulative incidence function (CIF) to estimate the probability of an event, accounting for competing risks. The 1-CIF, representing the probability of remaining event-free, was used to plot survival curves. Renal cancer-specific survival (CSS) was evaluated using a competing risks cause-specific hazard (Fine-Gray subdistribution hazard) model [28] controlling for Charlson comorbidity score (median), age, gender, the extent of disease

at diagnosis, modality of treatment, and period. The model also accounts for censoring among those who do not have an event during follow-up.

The proportionality assumption was checked visually for all time-to-event models.

Regression plots were used to demonstrate survival differences graphically. We performed subgroup analyses stratified by the time of RC diagnosis (before 2000, 2000–2004, 2005–2009, 2010–2014, and 2015–2017) to estimate changes in survival across the three decades.

All statistical p-values are 2-sided with a level of significance set at $P < 0.05$. All analyses were performed with the IBM SPSS Statistics version 27 and Stata v18.4.

## Results

### Study cohort

A total of 18,700 RC cases were identified from the Finnish Cancer Registry FCR. We excluded 1,613 cases where the first RC diagnosis was recorded post-mortem. After excluding 1,613 cases with post-mortem diagnosis, 182 cases with age < 18 or >100 years, and 2,492 cases with missing histological data, the final cohort consisted of 14,413 cases (Fig 1).

### Cancer cases diagnosed in Finland during 1995–2018

**Population characteristics.** The primary analysis cohort included 14,413 RC cases (median age 67 years, IQR 60.0–77.0, range 18–100) at diagnosis (Table 1). The majority were male (56.9%). The median follow-up time was 57.0 months (IQR 13.0–103.0). Clear cell RCC (ccRCC) was the most common histological subtype (75.5%), followed by papillary RCC (pRCC, 5.8%), chromophobe RCC (chRCC, 2.1%), collecting duct carcinoma (CDC, 0.1%), and sarcomatoid RCC (sarcRC, 0.6%). Other known RCC subtypes accounted for 1.2% of cases. (Table 1) During the study period, the proportion of diagnose non-clear cell renal cell carcinoma (non-ccRCC) cases increased significantly, from 2.2% of all cases in 1995–1999 to 19.8% in 2015–2017.

At the time of diagnosis, localized RCC was the most prevalent, accounting for 46.1% of cases, followed by metastatic disease 23.9% (3442 cases) and locally advanced RC (2.7%). Stage data was missing for 27.3% of cases. There was no

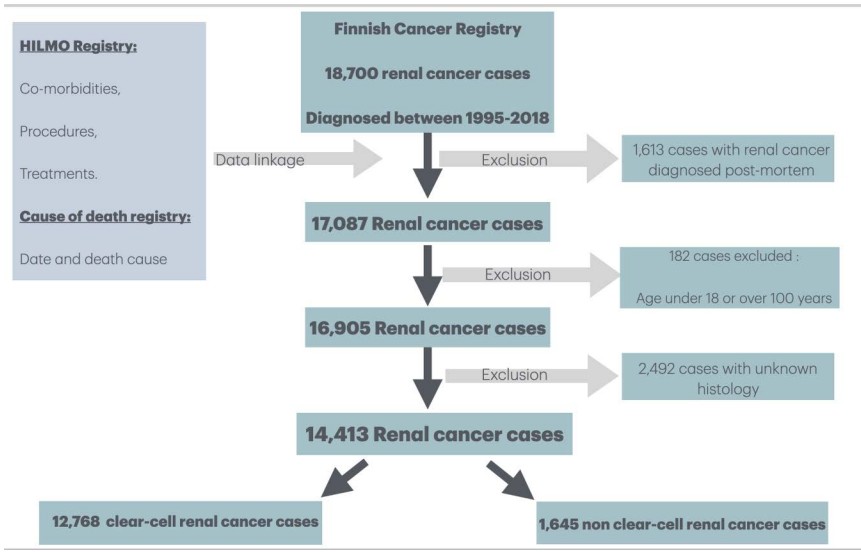

**Fig 1. Flow Chart for Formation of the Population-based Cohort of 14,593 Renal.**

**Table 1. Demographic, Clinical, Treatment Characteristics and Mortality Outcomes in Patients with Clear Cell and Non-Clear Cell Renal Cell Carcinomas in Finland, 1995–2017.**

| Characteristic | Clear cell RCC (n = 12,768) | Non-clear cell RCC (n = 1,645) | Papillary RCC (n = 978) | Chromophobe RCC (n = 352) | Sarcomatoid RC (n = 100) | Collecting duct carcinoma (n = 17) | Other known histological subtypes (n = 198) |
|---|---|---|---|---|---|---|---|
| **Follow-up (months)** | | | | | | | |
| Mean (SD) | 79.15 (69.65) | 60.9 (50.4) | 65.2 (52.3) | 59.9 (36.9) | 43.1 (61.9) | 22.5 (29.3) | 54.3 (52.7) |
| Median (IQR) | 59.0 (22.0, 120.0) | 49.0 (21.0, 89.0) | 51.0 (25.0, 93.0) | 55.0 (28.3, 87.0) | 13.5 (7.0, 46.3) | 9.0 (5.0, 29.5) | 41.0 (9.0, 87.3) |
| P-value | Reference | <0.001 | 0.002 | 0.035 | <0.001 | <0.001 | <0.001 |
| **Gender** | | | | | | | |
| Male (n, %) | 7076 (55.4%) | 1118 (68.0%) | 735 (75.2%) | 205 (58.2%) | 52 (52.0%) | 12 (70.6%) | 114 (57.6%) |
| Female (n, %) | 5691 (44.6%) | 527 (32.0%) | 243 (24.8%) | 147 (41.8%) | 48 (48.0%) | 5 (29.4%) | 84 (42.4%) |
| P-value | Reference | <0.001 | <0.001 | 0.295 | 0.493 | 0.209 | 0.546 |
| **Age at Diagnosis (years)** | | | | | | | |
| Median (IQR) | 67.0 (59.0, 75.0) | 67.0 (58.0, 75.0) | 66.0 (58.0, 74.0) | 68.0 (58.0, 75.0) | 66.5 (56.0, 74.0) | 69.0 (59.0, 79.0) | 68.5 (56.0, 76.3) |
| P-value | Reference | 0.077 | 0.017 | 0.807 | 0.285 | 0.452 | 0.754 |
| *Charlson Comorbidity Index* | | | | | | | |
| Median (IQR) | 2.8 (2.0, 3.0) | 2.9 (2.0, 3.0) | 2.9 (2.0, 3.0) | 2.8 (2.0, 3.0) | 3.0 (2.0, 3.0) | 2.9 (2.0, 3.5) | 3.1 (2.0, 3.0) |
| P-value | Reference | <0.001 | 0.001 | 0.162 | 0.201 | 0.258 | 0.163 |
| **Tumor Laterality (n, %)** | | | | | | | |
| Left | 5325 (41.7%) | 651 (39.6%) | 400 (40.9%) | 138 (39.2%) | 37 (37.0%) | 11 (64.7%) | 81 (40.9%) |
| Right | 5573 (43.6%) | 645 (39.2%) | 356 (36.4%) | 152 (43.2%) | 52 (52.0%) | 4 (26.7%) | 65 (32.8%) |
| Bilateral | 110 (0.9%) | 22 (1.3%) | 16 (1.6%) | 2 (0.6%) | 1 (1.0%) | 0 (0.0%) | 3 (1.5%) |
| Unknown | 1767 (13.8%) | 327 (19.9%) | 206 (21.1%) | 60 (17.0%) | 10 (10.0%) | 2 (11.8%) | 49 (24.7%) |
| P-value | Reference | 0.054 | 0.002 | 0.79 | 0.389 | 0.151 | 0.275 |
| **Tumor Extent at Diagnosis (n, %)** | | | | | | | |
| Local | 5991 (46.9%) | 655 (39.8%) | 424 (43.4%) | 142 (40.3%) | 21 (21.0%) | 3 (17.6%) | 65 (32.8%) |
| Locally Advanced | 332 (2.6%) | 54 (3.3%) | 25 (2.5%) | 19 (5.4%) | 6 (6.0%) | 0 (0.0%) | 5 (2.5%) |
| Metastatic | 3135 (24.6%) | 307 (18.7%) | 124 (12.7%) | 44 (12.5%) | 57 (57.0%) | 8 (47.1%) | 74 (37.4%) |
| Data Missing | 3310 (25.9%) | 629 (38.2%) | 406 (41.5%) | 147 (41.8%) | 16 (16.0%) | 6 (35.3%) | 54 (27.3%) |
| P-value | Reference | <0.001 | <0.001 | <0.001 | <0.001 | 0.041 | <0.001 |
| *Treatment (n, %)* | | | | | | | |
| Surgical | 8676 (68.0%) | 759 (46.1%) | 462 (47.2%) | 147 (41.8%) | 56 (56.0%) | 4 (26.7%) | 90 (45.5%) |
| P-value | Reference | <0.001 | <0.001 | <0.001 | 0.11 | <0.001 | <0.001 |
| Medical (TKI/mTOR)* | 632 (4.9%) | 39 (2.4%) | 24 (2.5%) | 5 (1.4%) | 7 (7.0%) | 1 (5.9%) | 2 (1.0%) |
| P-value (difference) | Reference | <0.001 | <0.001 | 0.002 | 0.347 | 0.859 | 0.011 |
| **Mortality Rate (per 10,000 person-years)** | | | | | | | |
| All-cause | 847.3 | 706.3 | 598.9 | 438.6 | 2198.5 | 4386.0 | 1138.2 |
| Renal cancer-specific | 481.8 | 348.4 | 265.5 | 125.3 | 1864.6 | 3132.8 | 569.1 |

*TKI/mTOR- Tyrosine Kinase Inhibitor/ mammalian target of Rapamycin.

significant difference in age or Charlson's comorbidity score between ccRCC and non-ccRCC. However, follow-up time was significantly longer for ccRCC (median 59.0 months) compared to non-ccRCC (median 49.0 months, p<0.001).

**Mortality.** During follow-up 7,136 subjects (55.9%) in the ccRCC group died; of these 4,058 (31.6%) died due to RC. In the non-ccRCC group, 590 subjects (35.9%) died; of these, 291 (17.6%) died due to RC.

Renal cancer-specific mortality was significantly higher for ccRCC (481.8 per 10,000 person-years) compared to non-ccRCC (348.4 per 10,000 person-years, p<0.0001). All-cause mortality was also higher for ccRCC (847.3 per 10,000 person-years) versus non-ccRCC (706.3 per 10,000 person-years, p=0.0003) (Table 1).

**Surgical treatment.** Surgery was the primary treatment modality in 9,435 cases (65.5%). The most common procedure was total nephrectomy (n=7,739, 82.0%). Partial nephrectomy was used in 841 cases. In 5.9% of the cases, data on the precise surgical method was missing.

Up to 31.1% of surgically-treated RCC cases died during the follow-up due to RC. Among surgically treated patients, mortality rates due to RC varied significantly (P<0.001) across histological subtypes: 6.8% for chRCC, 17.9% for pRCC, 32.1% for ccRCC, and with the highest rates observed for CDC and sarcRC at 50.0% and 62.5%, respectively.

### Overall and cancer-specific survival by histological RCC subtype

*Overall* **survival.** Overall survival (OS) differed significantly between histological subtypes. In univariate analysis, non-ccRCC demonstrated better OS (5-year survival (5-ySR) 72.6%, 95% CI 70.3–74.7%) than ccRCC (5-ySR 62.7%, 95%CI 61.8–63.5). (Fig 2) However, the OS differs significantly among the different non-ccRCC subtypes. The 5-ySR for pRCC is 74.3%(95% CI 71.2–77.2), and for chRCC 82.2% (95% CI 77.2–86.1) is better compared to ccRCC. Other known variants appear to have similar OS to ccRCC 5-ySR 72.9% (95% CI 67.9–77.2). In contrast sarcRC and CDC had significantly lower OS, with 5-ySR 29.2% (95% CI 20.6–38.3) and 23.5% (95%CI 7.3–44.9) (Fig 3).

In adjusted Cox regression analysis, the general pattern remains the same: non-ccRCC had better OS (HR 0.87, 95%CI 0.79–0.94) compared to ccRCC (Table 2). Similar results for survival were observed when the non-ccRCC subgroup was divided into subgroups. (Table 2)

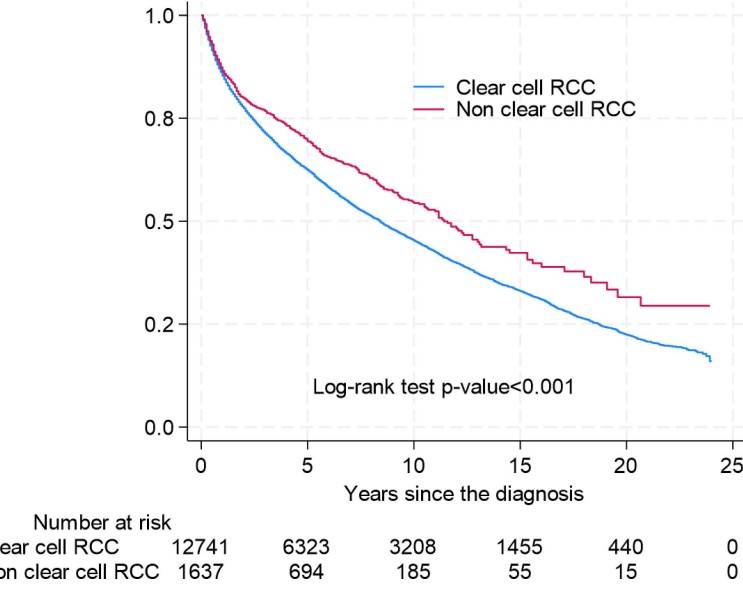

| Number at risk | | | | | | |
|---|---|---|---|---|---|---|
| Clear cell RCC | 12741 | 6323 | 3208 | 1455 | 440 | 0 |
| Non clear cell RCC | 1637 | 694 | 185 | 55 | 15 | 0 |

**Fig 2. Kaplan-Meier Curves for Overall Survival for ccRCC and non-ccRCC Subgroups combined.**

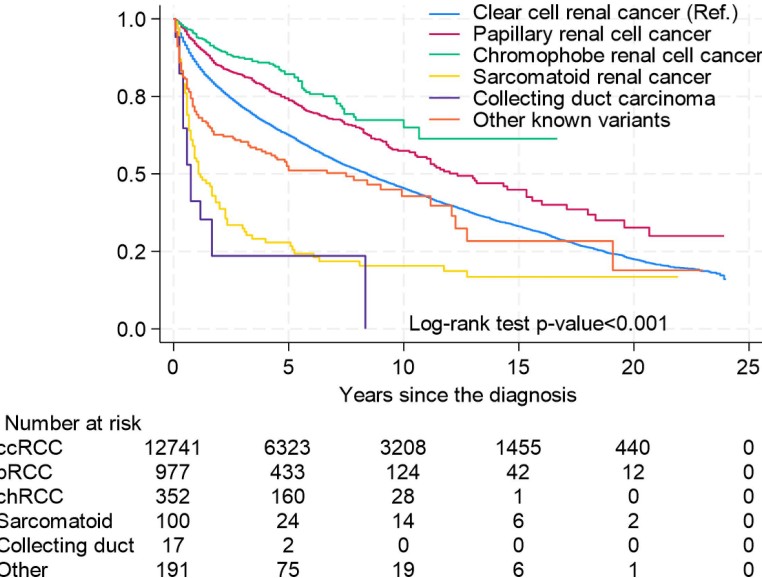

| Number at risk | | | | | |
|---|---|---|---|---|---|
| ccRCC | 12741 | 6323 | 3208 | 1455 | 440 | 0 |
| pRCC | 977 | 433 | 124 | 42 | 12 | 0 |
| chRCC | 352 | 160 | 28 | 1 | 0 | 0 |
| Sarcomatoid | 100 | 24 | 14 | 6 | 2 | 0 |
| Collecting duct | 17 | 2 | 0 | 0 | 0 | 0 |
| Other | 191 | 75 | 19 | 6 | 1 | 0 |

**Fig 3. Kaplan-Meier Curves for Overall Survival for ccRCC and non-ccRCC Subgroups Separated.**

**Table 2. Overall and renal cancer-specific survival outcomes in clear-cell renal cell cancer vs non-clear cell renal cell carcinoma (combined and in subgroup separated).**

| Characteristic | Overall Survival[1] | | Renal Cancer–Specific Survival[3] | |
|---|---|---|---|---|
|  | Adj HR[2] 95%CI | p-value | Adj SHR[4] 95%CI | p-value |
| **A. Overall Comparison of Histological Subtypes** | | | | |
| Clear cell RCC | Reference | | Reference | |
| non-ccRCC group combined | 0.87 (0.79-0.94) | 0.001 | 0.69 (0.60-0.78) | < 0.001 |
| **B. Comparison of Non-Clear Cell Cancer Subtypes** | | | | |
| Clear cell RCC | Reference | | Reference | |
| Papillary RCC | 0.80 (0.71-0.90) | < 0.001 | 0.66 (0.56-0.78) | < 0.001 |
| Chromophobe RCC | 0.56 (0.45-0.71) | < 0.001 | 0.28 (0.18-0.43) | < 0.001 |
| Sarcomatoid renal cancer | 2.03 (1.62-2.53) | < 0.001 | 1.83 (1.36-2.46) | < 0.001 |
| Collecting duct carcinoma | 3.65 (2.16-6.19) | < 0.001 | 3.19 (2.01-5.08) | < 0.001 |
| Other known variants | 0.95 (0.78-1.16) | 0.612 | 0.53 (0.38-0.77) | 0.001 |

[1]Multivariable Cox proportional hazards regression model (adjusted for gender, histology, age, Charlson Comorbidity Index, treatment, tumor extent, and time of diagnosis).

[2]Adjusted hazard ratio.

[3]Multivariable Fine and Grays proportional subhazards model (adjusted for gender, histology, age, Charlson Comorbidity Index, treatment, tumor extent, and time of diagnosis).

[4]Adjusted subdistribution hazard ratio

**Renal cancer-specific survival.** In CSS, we see a similar pattern emerging to that of univariate analysis. Non-ccRCC as a combined group [5-year cancer-specific survival rate (5-yCSSR) 84.6%, 95% CI 82.8–86.2] has better survival outcomes compared to ccRCC (5-yCSSR 73.5%, 95% CI 72.7–74.3) (Fig 4).

When analyzing non-ccRCC group histological subtypes separately, we see again the pattern of CSS being better in general for chRCC (5-yCSSR 94.1%, 95% CI 91.2–96.3), papillary RCC (5y-CSSR 86.8%, 95% CI 84.4–88.9) and

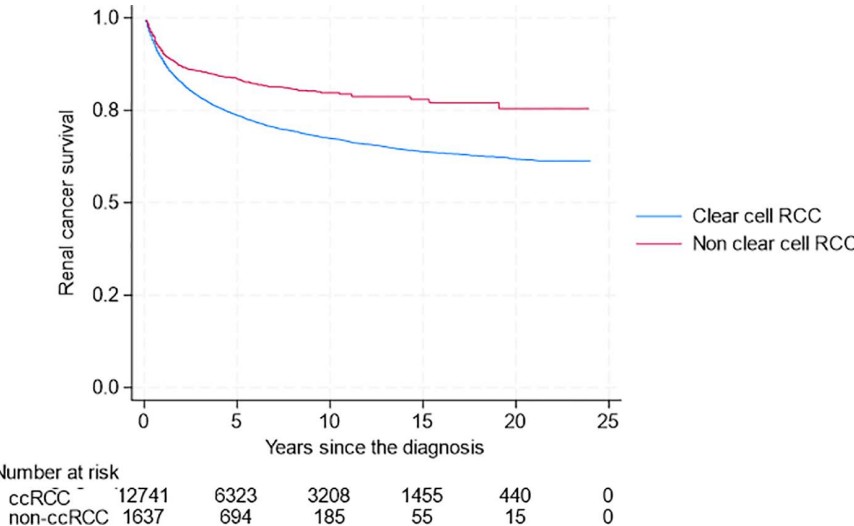

**Fig 4. Renal Cancer-Specific Survival for ccRCC and non-ccRCC Subgroups Combined.**

otherRC (5-yCSSR 75.4%, 95% CI 69.0–81.3) compared to ccRCC (5-yCSSR 73.5%, 95% CI 72.7–74.3). Whereas having a sarcomatoid differentiation (5yCSSR 37.1%, 95% CI 28.4–47.5) or CDC is related to inferior survival rates (5-yCSSR 44.1%, 95% CI 24.6–70.0) (Fig 5).

The Fine Gray competing risks model results were consistent with the findings from the overall survival model, we observed that non-clear cell renal cell carcinoma (non-ccRCC) was associated with improved cancer-specific survival (CSS) compared to clear cell renal cell carcinoma (ccRCC) (sHR 0.69, 95% CI 0.60–0.78) (Table 2).

Furthermore, specific subtypes of non-ccRCC were linked again to more favorable CSS outcomes relative to ccRCC: chRCC demonstrated a sHR of 0.28, 95% CI 0.18–0.43), pRCC exhibited a sHR of 0.66, 95% CI 0.56–0.78, and otherRC had a sHR of 0.53, 95% CI 0.38–0.77 (Table 2). In contrast, specific subtypes were associated with poorer CSS, such as sarcomatoid variants (sHR 1.83, 95% CI 1.36–2.46) and collecting duct carcinoma (CDC) (sHR 3.19, 95% CI 2.01–5.08) (Table 2).

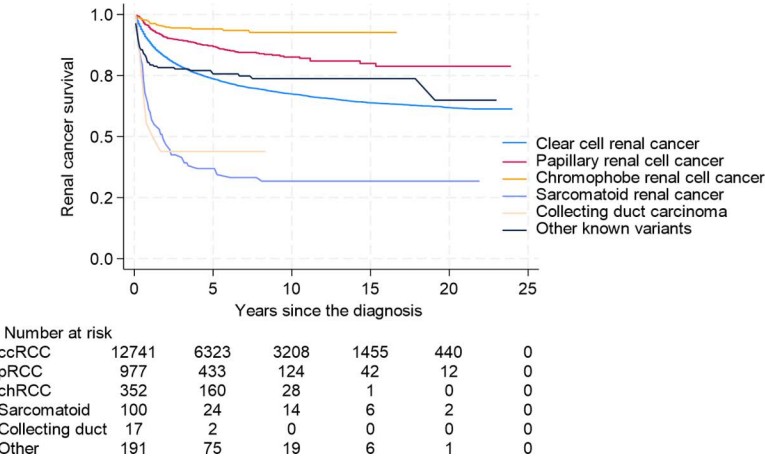

**Fig 5. Renal Cancer-Specific Survival for ccRCC and non-ccRCC Subgroups Separated.**

**Impact of surgical treatment on different histological subtypes.** Multivariable Cox regression analysis on cancer-specific survival revealed that the survival benefit of surgery was most pronounced in patients with ccRCC compared to conservative management (aHR 2.19, 95% CI 2.01–2.37). Although a trend towards improved survival with surgery was observed in non-ccRCC subtypes combined (aHR1.29, 95% CI 0.94–1.76) and as separate entities, these associations did not reach statistical significance.

Comparison of conservative treatment with total and partial nephrectomy demonstrated that patients undergoing partial nephrectomy achieved superior survival outcomes. Moreover, the findings reaffirmed that surgical intervention provided the greatest benefit for patients with ccRCC (Table 3).

As this study is retrospective, we can expect a clinician bias in allocating patients for their preferred management.

**RCC survival trend over the years.** *R*C cancer-specific survival has dramatically improved over the decades. Here, we divided RCC cases into yearly groups of 1995–2000, 2000–2004 (sHR 1.0, 95% CI 0.93–1.09), 2005–2009 (sHR 0.75, 95% CI 0.70–0.82), 2010–2014 (sHR 0.50, 95% CI 0.46–0.54), and 2015–2017 (sHR 0.38, 95% CI 0.34–0.42). In general, survival improved from the 1990s to 2010–2014. Our analysis showed a 62% death risk reduction from the start of the study period (1995–1999) compared to 2015–2017 (Fig 6.)

In dividing up different histological subtypes, we can see cancer-specific survival improvement over the years, but with some differences. As most cases in our cohort are ccRCC, then analyzing the ccRCC subgroup separately (Fig 7) does not differ significantly from the analysis consisting of all RC cases (Fig 6). The pattern shows a comparable level of improvement, with the years 1995−2004 being nearly indistinguishable, while the highest survival rates are noted in 2015−2017 (SHR 0.44, 95% CI 0.38–0.51), indicating a 56% lower risk of dying from renal cancer.

When examining the combined subtypes group of chRCC, pRCC, and other known RC separately, the group does not follow the typical pattern of improvement over the years. The cohort from 2000–2004 exhibited the poorest survival rates (SHR 1.31, 95% CI 0.63–2.73), while patients grouped from 2010–2014 had the best survival outcomes (SHR 0.70, 95% CI 0.37–1.31) (Fig 8).

However, a low number of cases still limits the analysis for these subgroups and if we group all non-ccRCC in periods of 1995–2004 and 2005–2017, the risk of death has improved by 36% (Fig 9.)

## Discussion and conclusions

### Renal cell carcinoma: heterogeneity

Optimal treatment decisions for renal cell carcinoma (RCC) must incorporate clinical and pathological findings. This large-scale, whole population-based study provides a comprehensive analysis of RCC clinical features and prognosis, focusing

**Table 3. Renal cancer-specific survival outcomes among partial and total nephrectomy in reference to non-surgical treatment cases.**

|  | Total nephrectomy | | Partial nephrectomy | |
|---|---|---|---|---|
|  | **Adj. HR, 95% CI** | **p-value** | **Adj. HR, 95% CI** | **p-value** |
| Clear cell RCC | 0.48 (0.44-0.53) | <0.001 | 0.16 (0.11-0.21) | <0.001 |
| Non-clear cell RCC | 0.84 (0.61-1.15) | 0.28 | 0.27 (0.12-0.61) | 0.002 |
| Papillary RCC | 0.94 (0.56-1.58) | 0.81 | 0.31 (0.13-0.75) | 0.009 |
| Chromophobe RCC | 0.48 (0.09-2.54) | 0.39 | *1 | 0.98 |
| Sarcomatoid RC | 0.52 (0.29-0.96) | 0.36 |  |  |
| Collecting duct carcinoma | 0.11 (0.01-2.63) | 0.17 |  |  |
| Other known variants | 0.81 (0.39-1.66) | 0.56 | *1 |  |

*1- The sample size was too small for meaningful statistical analysis.

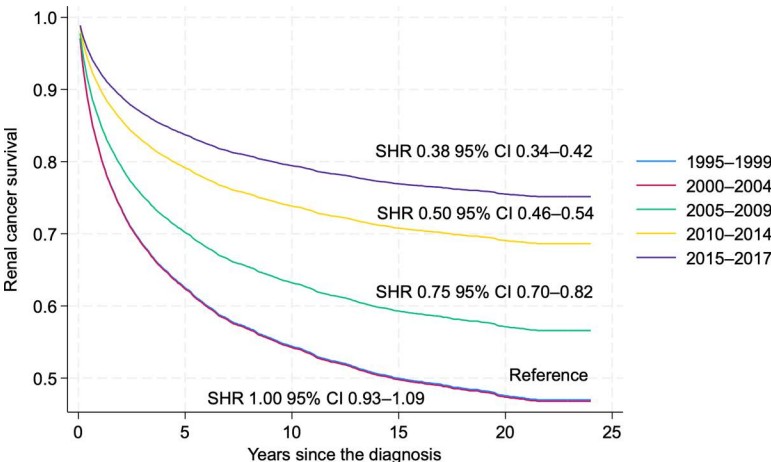

**Fig 6. Renal Cancer (all types combined) Cancer-Specific Survival by Period (1995-1999, 2000–2004, 2005–2009, 2010–2014, 2015–2017) (reference – period 1995–1999).**

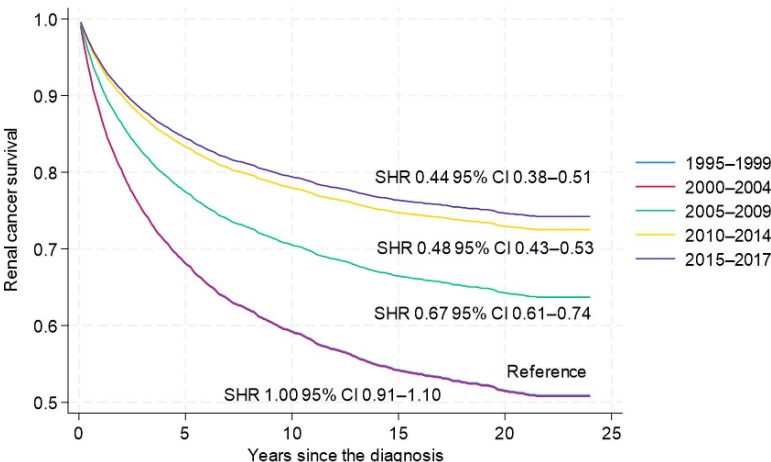

**Fig 7. Clear-cell Renal Cancer Cancer–Specific Survival by Period (1995-1999, 2000-2004, 2005–2009, 2010–2014, 2015–2017) (reference – period 1995–1999).**

on histological subtypes, and is not limited to only surgically treated patients. Our findings reveal substantial heterogeneity in cancer-specific survival among different RC subtypes, even after adjusting for factors such as tumor extent, age, and surgical treatment.

## Survival outcomes and tumor growth rates

While papillary RCC (pRCC) and chromophobe RCC (chRCC) demonstrated the best CSS, sarcomatoid RCC (sarcRC) and collecting duct carcinoma (CDC) had the poorest outcomes. This variation in survival may be partly attributed to differences in tumor growth rates. A previous study indicated a significantly slower growth rate for pRCC compared to ccRCC (0.017 cm vs. 0.28 cm per year [29].

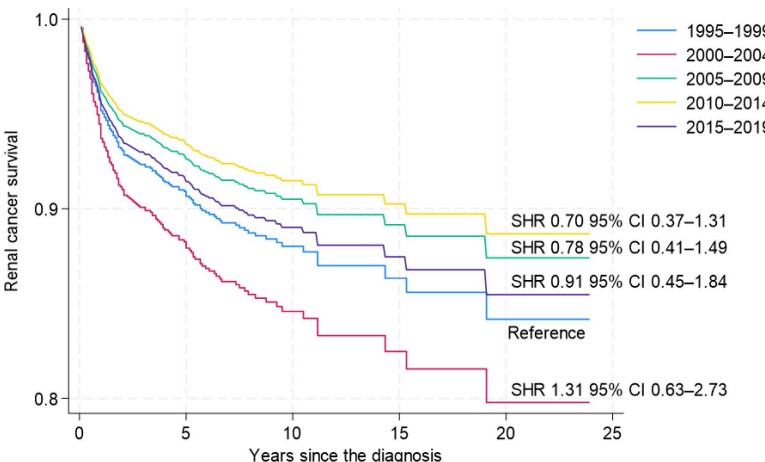

**Fig 8. Non-clear-cell Renal Cancer (combined of chromophobe RCC, papillary RCC andother known variants) Cancer-Specific Survival by Period (1995–2004, 2005–2017) (reference – period 1995–1999).**

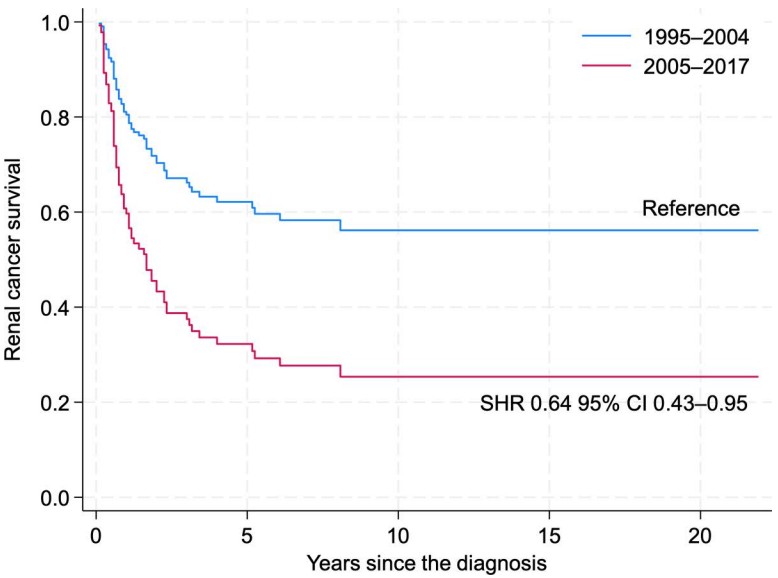

**Fig 9. Non-clear-cell Renal Cancer (all variants combined) Cancer-Specific Survival by Period (1995–2004, 2005–2017) (reference – period 1995–1999).**

## Stage migration and its impact

We observed improved RCC survival over time, with patients diagnosed after 2000 having better prognoses than those diagnosed in the 1990s. This likely reflects stage migration towards a more localized RC due to the increased usage of imaging and more efficient treatment modalities. A study in East of England observed a 16% increase in stage I from 1999–2003 vs 2014–2016), with a 11% decrease in the same periods in stage IV (or missing stage) [12].

This phenomenon, known as stage downshift, has been widely reported in the literature. [11,12,30–34].

Importantly, research has shown a differential impact of stage shift on clear cell renal cell carcinoma (ccRCC) and non-clear cell renal cell carcinoma (non-ccRCC) [32–34]. Also, our data support that stage downshift is more pronounced for ccRCC than for other subtypes. Notably, non-ccRCC diagnosed after 2000 did not necessarily have better survival, with pRCC, chRCC, and other known RC had the poorest survival outcomes in 2000–2004. In non-ccRCC, stage shift was associated with a decrease in tumor size and a trend towards lower grades. Conversely, in ccRCC stage shift was associated with a shift toward more aggressive grades (G3, G4). Further, node-positive and metastatic disease decreased for non-ccRCC, while remaining stable for ccRCC. These findings support the concept that stage downshift has a varying effect depending on the histological subtype of RCC [10].

## Surgical management and treatment considerations

Surgery (total or partial nephrectomy) is currently the primary treatment option in localized or locally advanced RC [32]. While we observed survival benefits for all surgically treated subtypes, this benefit was statistically significant only for ccRCC. This may be partly due to the lower proportion of surgically managed cases among the non-ccRCC group, resulting in limited statistical power to evaluate the benefit of surgery. Notably, chRCC and pRCC (histological subtypes characterized by good prognosis) demonstrated excellent CSS regardless of primary treatment, possibly due to overdiagnosis and, thereafter, overtreatment [35].

## Minimally invasive and conservative approaches

The role of less invasive or conservative methods, such as thermal ablation or active surveillance (AS), is increasingly being considered for smaller tumors [36]. Tumor ablation techniques, including cryoablation, microwave ablation, and radiofrequency ablation, have demonstrated promising short-term outcomes, characterized by high efficacy and a low incidence of complications [37].

Our findings support using more conservative treatment for pRCC, chRCC and other known subtypes as their CSS was not significantly impacted by surgery or earlier diagnosis. Accumulating evidence suggests similar CSS outcomes for surgery, ablation therapy (TA), or (AS). For patients over 75 years of age, comorbidities may be more critical in determining outcomes, as surgery can have negative cardiovascular effects [32–34]. Studies have shown that surgical interventions alone can increase the risk of myocardial infarction and stroke [38]. Undergoing a partial or total nephrectomy results in the loss of renal tissue, which may later be associated with chronic kidney damage (CKD.) The risk of CKD is particularly significant among patients who undergo total nephrectomy [39]. Furthermore, post- surgical acute kidney failure has been associated with an increased likelihood of developing CKD, leading to greater morbidity and mortality risks in the future [40].

The potentially limited benefit of surgery for non-clear cell renal cell carcinoma (non-ccRCC) may stem from their naturally slower growth rates [35,41]. As a result, patients with non-ccRCC are more likely to die from other health issues before their kidney tumors become life-threatening.

## Histological confirmation and future directions

Although more data is evidently needed, our study suggests that non-ccRCC prognosis is less influenced by treatment or diagnostic advancements compared to ccRCC. These tumors may grow slower, limiting the potential benefits of surgery. Accurate histological confirmation of tumor histology is crucial for personalized treatment planning. RC biopsies were underutilized during our study period (only 789 cases (4.2%)), and of these biopsied 17.1% were described as having metastatic disease.

Percutaneous RC biopsy has demonstrated high accuracy (90%−92%) with low complication rates [42,43]. A long-lasting myth about tumor seeding has been refuted [44]. This raises the question of whether histological proof should be considered a routine part of the decision process before initiating any other treatment intervention. Further research is warranted to evaluate the potential benefits and implications of incorporating biopsies into the standard RC management pathway.

## Study strengths and limitations

The study's main strength was the use of nationwide population-based data from a high-quality and validated cancer registry. The adequately large sample size allowed us to provide further knowledge and data on the rare RC types. Follow-up covers several decades, providing sufficient power to analyze long-term trends in survival. Our study includes in-depth information on comorbidities.

However, limitations include missing data on tumor TNM stage, exact tumor size, ISUP grade and the broad definition of surgical treatment may have led to underreporting of partial nephrectomies.

In our analysis, 2,492 cancer cases with missing histology information were excluded. The significantly shorter follow-up time in the excluded group suggests that these patients may have had a more aggressive disease course or experienced earlier mortality. The older age and higher proportion of metastatic disease further support this possibility. Consequently, our analysis may have underestimated the impact of these factors on overall survival and cancer-specific survival. The Finnish population is predominantly of Caucasian origin, so extrapolating our results to other ethnicities should be considered with caution [45]. Several studies have demonstrated that renal cell carcinoma survival rates among Black individuals are worse compared to their White counterparts [46,47]. However, research indicates that this survival difference could be primarily driven by racial disparities in comorbidities, such as chronic renal failure, and socioeconomic deprivation. When adjusted for socioeconomic factors, the survival difference was no longer statistically significant [46]. Further, the proportional hazards assumption in survival analysis was assessed visually. While the assumption appeared to hold for the primary analyses, some degree of non-proportionality may be present. It is important to note that proportional hazards (PH) violation does not invariably lead to biased hazard ratio estimates. When censoring is absent or independent of covariates, average hazard ratios retain their validity [48].

## Conclusion

This study highlights diverse survival trends across RCC histological types. Non-ccRCC are not uniform in terms of prognosis. Some non-ccRCC-s (pRCC,chRCC and otherRC) have significantly better prognoses compared to ccRCC, while some (sarcomatoid RC and CDC) have worse prognoses. As the incidence of renal cell carcinoma (RC) remains high, treatment strategies should be tailored to individual subtype and patient characteristics. Surgery plays a more significant role in clear cell renal cell carcinoma (ccRCC), particularly when detected early, as early detection allows for more effective surgical intervention. Minimally invasive treatments, such as thermal ablation or active surveillance, may be more suitable for subtypes with good prognoses.

## Author contributions

**Conceptualization:** Teesi Sepp, Andres Kotsar, Thea Veitonmäki.

**Data curation:** Teesi Sepp, Antti Poyhonen, Anneli Uusküla, Teuvo L. J. Tammela, Aleksei Baburin, Teemu J. Murtola.

**Formal analysis:** Teesi Sepp.

**Methodology:** Teesi Sepp, Antti Poyhonen, Anneli Uusküla, Aleksei Baburin, Teemu J. Murtola.

**Project administration:** Teesi Sepp, Andres Kotsar.

**Supervision:** Anneli Uusküla, Andres Kotsar, Teemu J. Murtola.

**Visualization:** Teesi Sepp, Aleksei Baburin.

**Writing – original draft:** Teesi Sepp.

**Writing – review & editing:** Teesi Sepp, Anneli Uusküla, Thea Veitonmäki, Teemu J. Murtola.

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
