## [Decision Letter · Decision Letter 0]

Dear Dr. Sepp,

Thank you for submitting your manuscript to PLOS ONE. After careful consideration, we feel that it has merit but does not fully meet PLOS ONE’s publication criteria as it currently stands. Therefore, we invite you to submit a revised version of the manuscript that addresses the points raised during the review process.

We look forward to receiving your revised manuscript.

Kind regards,

Yuki Arita, M.D., Ph.D

Academic Editor

PLOS ONE

Journal Requirements:

None

5. Please amend your list of authors on the manuscript to ensure that each author is linked to an affiliation. Authors’ affiliations should reflect the institution where the work was done (if authors moved subsequently, you can also list the new affiliation stating “current affiliation:….” as necessary).

Additional Editor Comments :

This manuscript presents an intriguing and relevant study with significant potential to contribute to its field.

The topic is timely, and the research question is well-defined, addressing an area of growing interest. The authors are commended for their effort in designing and conducting this study.

However, after a thorough review, I believe that some substantial revisions are necessary to improve the clarity, robustness, and overall impact of the work.

Several sections require additional detail to support the study's methodology and findings fully.

There are also areas where the presentation of data and discussion could be strengthened to better highlight the study's implications and relevance in the broader context.

Reviewers' comments:

Reviewer's Responses to Questions

**Comments to the Author**

1. Is the manuscript technically sound, and do the data support the conclusions?

Reviewer #1: Yes

Reviewer #2: Yes

Reviewer #3: Yes

2. Has the statistical analysis been performed appropriately and rigorously?

Reviewer #1: Yes

Reviewer #2: Yes

Reviewer #3: No

3. Have the authors made all data underlying the findings in their manuscript fully available?

Reviewer #1: Yes

Reviewer #2: No

Reviewer #3: No

4. Is the manuscript presented in an intelligible fashion and written in standard English?

Reviewer #1: Yes

Reviewer #2: Yes

Reviewer #3: Yes

Reviewer #1: This is a well written article which provides a comprehensive analysis of renal cancer outcomes. It provides valuable insights into the prognostic differences between ccRCC and non-ccRCC histological subtypes. It concludes that Non -clear cell RCC patients generally has better OS and CSS compared to Clear cell RCC. It acknowledges that there has been significant advances in Imaging and treatment which has resulted in improvement in survival rates. There are several areas where the article can be improved

Elaborate further during discussion on why surgical treatment - Nephrectomy is beneficial in CC RCC but not so much in Non CC RCC ?

Elaborate further during discussion about Non surgical Management ?

Very few patients of Non CC RCC were included in the later years cohort and should be explained if that had any affect in results?

Studies included only caucasians which is mentioned in limitations and so limits generalizability. Could mention if there are biological differences between different ethnicities which could impact survival and treatment responses

Provide some possible clinical recommendations ,particularly how these findings can influence management for patients with different subtypes of RCC?

Reviewer #2: Title: Renal cancer survival in clear cell renal cancer compared to other types of tumor histology: a population-based cohort study

Abstract: Adequately written. Will benefit from a professional proofreading, study objective was unclear in the abstract, methodology & results were adequate. Conclusion can be improved further by indicating how the findings could affect patient care.

Introduction: Largely adequate.Suggest getting a proofreader to check on punctuation, decimals(e.g., line 54: 5.8/100,000 and line 57: 18.57)

Methodologies: Technically sound but the writing could benefit from a bit of proofreading.

Results:

Table1: some data are obscured. Suggest to provide a table with clearer data. Suggest to convert table to pdf version before publication.

Mortality analysis: It would be good to add multivariate analysis that could calculate the cancer specific mortality risk for someone with ccRCC compared to those diagnosed with non-ccRCC

Overall survival

Line 220: non-ccRCC demonstrated better OS than ccRCC (5-year survival (5-ySR) 72.6%, 95% 70.3-74.7%

- the result writing for non-ccRCC should be looked at again

Table3: the table provided is still raw. Suggest to polish the table for publication. The surgical treatment data could not account for the various indications for surgery or if the tumour was inoperable, so this should be mentioned as a limitation. Also, it would be good to compare the survival outcomes for patients receiving total vs partial nephrectomy if these data are available.

Discussions: Largely adequate.

Explanation that support stage downshift will improve the discussion further. Add more references that cover this aspect to strengthen the discussion.

Expansion on other non-surgical techniques with additional references related to this would improve the discussion further.

References:

Inadequate numbers of references cited for this manuscript.

Reviewer #3: please address these concerns in your revision:

Statistical Model Assumptions: Indicate if the Cox and Fine-Gray models' proportional hazards assumption was examined. If breached, take into account different modeling techniques.

Handling Missing Data: Clearly explain how missing data was handled, especially for histology (13%), and tumor extent (27.3%). If imputation was applied, describe the process. If left out, talk about possible bias.

Histological Classification Over Time: Explain whether or not older cases were reclassified using newer histological standards and how this could affect the results.

Kaplan-Meier Analysis: To verify the statistical significance of survival differences, include log-rank p-values in Kaplan-Meier curves.

Good luck.

**Do you want your identity to be public for this peer review?** For information about this choice, including consent withdrawal, please see our Privacy Policy

Reviewer #1: No

Reviewer #2: **Yes: ** INTAN SUHANA ZULKAFLI

Reviewer #3: **Yes: ** Mohammad Reza Fattahi

---

## [Author Response · Author response to Decision Letter 1]

1 Jul 2025

Dear Dr Yuki Arita, M.D., Ph.D.

We are pleased to submit a revised draft of our manuscript titled “Renal cancer survival in clear cell renal cancer compared to other types of tumor histology: a population-based cohort study” to PLOS ONE.

We appreciate the time and effort that you and the reviewers dedicated to providing constructive feedback on our initial submission. We believe we have successfully addressed all of the concerns raised in the review, and that the manuscript has been strengthened as a result.

Our responses to the review comments are summarized below.

Thank you for your consideration of this manuscript.

We look forward to your decision and are happy to address any further questions or comments.

Sincerely,

Teesi Sepp

Response to Reviewers

EDITORIAL COMMENTS

Please ensure that your manuscript meets PLOS ONE's style requirements, including those for file naming. The PLOS ONE style templates can be found Done

Please complete your Competing Interests on the online submission form to state any Competing Interests. The authors have declared that no competing interests exist.

We note that you have indicated that there are restrictions to data sharing for this study. PLOS only allows data to be available upon request if there are legal or ethical restrictions on sharing data publicly. There are legal restrictions on sharing a de-identified data set according to the Finnish law 488/1999 on Medical Research. Access to data can be applied at https://findata.fi/en/

Please provide a complete Data Availability Statement in the submission form, ensuring you include all necessary access information or a reason for why you are unable to make your data freely accessible The data used in this study are not publicly available because they include personal-level information, and specific permission from the national authority FINDATA is required to access them.

Please amend your list of authors on the manuscript to ensure that each author is linked to an affiliation. Authors’ affiliations should reflect the institution where the work was done Done

However, after a thorough review, I believe that some substantial revisions are necessary to improve the clarity, robustness, and overall impact of the work.

Several sections require additional detail to support the study's methodology and findings fully.

There are also areas where the presentation of data and discussion could be strengthened to better highlight the study's implications and relevance in the broader context. We undertook a thorough revision of the manuscript to enhance its clarity and overall impact.

In the methodology section, we clarified that no data reclassification or imputation for missing data was performed, and we detailed the approach used to verify the proportionality assumption.

Additional findings were included to elaborate on the changes in the proportion of non-ccRCC cases observed throughout the study period. All tables were reformatted to improve clarity, with Table 3 being replaced, as requested by reviewers, with a more comprehensive table comparing partial and total nephrectomy to non-surgical management.

In the discussion section, readability was improved through the incorporation of subsection headings. We expanded on how varying subtype growth rates influence survival outcomes, provided an in-depth discussion on the differential impact of surgical treatment on ccRCC versus non-ccRCC, and included additional data on minimally invasive techniques for renal cancer. Histological confirmation was highlighted as a possible recommendation for incorporation into routine treatment protocols. Furthermore, new references were added to contextualize the findings within the broader literature, particularly regarding renal cancer stage shifts

The limitations section was revised to address the potential impact of ethnicity on our findings and to explain the rationale for excluding data on unknown histological subtypes. The conclusion was enhanced to emphasize key recommendations for potential updates to clinical guidelines.

REVIEWER 1

Elaborate further during discussion on why surgical treatment - Nephrectomy is beneficial in CC RCC but not so much in Non CC RCC? In the revised manuscript, we will expand the discussion to provide a more detailed explanation on this.

Specifically,

On page 16-17 lines 682- 703: “While papillary RCC (pRCC) and chromophobe RCC (chRCC) demonstrated the best CSS, sarcomatoid RCC (sarcRC) and collecting duct carcinoma (CDC) had the poorest outcomes. This variation in survival may be partly attributed to differences in tumor growth rates.A previous study indicated a significantly slower growth rate for pRCC compared to ccRCC (0.017cm vs. 0.28 cm per year. [29]“

On page 17-18, lines 724-952: “While we observed survival benefits for all surgically treated subtypes, this benefit was statistically significant only for ccRCC. This may be partly due to the lower proportion of surgically managed cases among the non-ccRCC group, resulting in limited statistical power to evaluate the benefit of surgery. Notably, chRCC and pRCC (histological subtypes characterized by good prognosis) demonstrated excellent CSS regardless of primary treatment, possibly due to overdiagnosis and, thereafter, overtreatment.[35]”

And to conclude this topic, we added on page18, lines 957-959: “The potentially limited benefit of surgery for non-clear cell renal cell carcinoma (non-ccRCC) may stem from their naturally slower growth rates [35,41]. As a result, patients with non-ccRCC are more likely to die from other health issues before their kidney tumors become life-threatening.”

Elaborate further during discussion about Non surgical Management ? We appreciate the reviewer's request to elaborate further on non-surgical management.

We now have written page 18, lines 941-959 the following:

Minimally invasive and conservative approaches

The role of less invasive or conservative methods, such as thermal ablation or active surveillance (AS), is increasingly being considered for smaller tumors.[36] Tumor ablation techniques, including cryoablation, microwave ablation, and radiofrequency ablation, have demonstrated promising short-term outcomes, characterized by high efficacy and a low incidence of complications.[37]

Our findings support using more conservative treatment for pRCC, chRCC and other known subtypes as their CSS was not significantly impacted by surgery or earlier diagnosis. Accumulating evidence suggests similar CSS outcomes for surgery, ablation therapy (TA), or (AS). For patients over 75 years of age, comorbidities may be more critical in determining outcomes, as surgery can have negative cardiovascular effects. [32 -34] Studies have shown that surgical interventions alone can increase the risk of myocardial infarction and stroke.[38] Undergoing a partial or total nephrectomy results in the loss of renal tissue, which may later be associated with chronic kidney damage (CKD.) The risk of CKD is particularly significant among patients who undergo total nephrectomy.[39] Furthermore, post- surgical acute kidney failure has been associated with an increased likelihood of developing CKD, leading to greater morbidity and mortality risks in the future.[40]

The potentially limited benefit of surgery for non-clear cell renal cell carcinoma (non-ccRCC) may stem from their naturally slower growth rates. [35,41] As a result, patients with non-ccRCC are more likely to die from other health issues before their kidney tumors become life-threatening.

Very few patients of Non CC RCC were included in the later years cohort and should be explained if that had any affect in results? We appreciate the reviewer's observation regarding the number of non-clear cell renal cell carcinoma (non-ccRCC) cases in later year cohorts. We understand the concern that a seemingly low number of these cases might have affected our results.

However, our analysis revealed a progressive increase in the proportion of non-ccRCC diagnoses over time. For instance, papillary renal cell carcinoma (pRCC) diagnoses increased from 1.43% in 1995-1999 to 12.3% in the 2015-2017 period. Similarly, chromophobe renal cell carcinoma (chRCC) diagnoses increased from 0% to 4.8% in the same time frame. (see the Table 1 below)

As for the manuscript, we added on page 9, lines 400-402 :" During the study period, the proportion of diagnose non-clear cell renal cell carcinoma (non-ccRCC) cases increased significantly, from 2.2% of all cases in 1995–1999 to 19.8% in 2015–2017.

Studies included only caucasians which is mentioned in limitations and so limits generalizability. Could mention if there are biological differences between different ethnicities which could impact survival and treatment responses The reviewer raises a crucial point about potential biological differences between ethnicities that could impact survival and treatment responses. While our study focused specifically on Caucasians, it is well-documented that variations in genetic predispositions, metabolic pathways, and tumor biology can exist across different ethnic groups.

To further emphasize the potential impact of ethnicity, we have added the following information to the article (pages 19-20 lines 1293-1366):” The Finnish population is predominantly of Caucasian origin, so extrapolating our results to other ethnicities should be considered with caution.[45] Several studies have demonstrated that renal cell carcinoma survival rates among Black individuals are worse compared to their White counterparts. [46-47] However, research indicates that this survival difference could be primarily driven by racial disparities in comorbidities, such as chronic renal failure, and socioeconomic deprivation. When adjusted for socioeconomic factors, the survival difference was no longer statistically significant.[46]“

Provide some possible clinical recommendations, particularly how these findings can influence management for patients with different subtypes of RCC? We have gladly added a few ideas, on how management of renal cancer could be improved and what could possibly benefit RC patients.

On page the 18, lines 955-956 we now have: “Our findings support using more conservative treatment for pRCC, chRCC and other known subtypes as their CSS was not significantly impacted by surgery or earlier diagnosis.”

And on the page 19 lines 1276-1280: „ This raises the question of whether histological proof should be considered a routine part of the decision process before initiating any other treatment intervention. Further research is warranted to evaluate the potential benefits and implications of incorporating biopsies into the standard RC management pathway..“

REVIEWER 2

Will benefit from a professional proofreading, study objective was unclear in the abstract, methodology & results were adequate. Conclusion can be improved further by indicating how the findings could affect patient care. We appreciate the input regards making the text better to read and have asked for additional proofreading and hopefully it deems sufficient.

In the Conclusion, we have made changes to emphasize the need for histological confirmation and histology-based treatment planning (page 20, lines 1375-1379):” As the incidence of renal cell carcinoma (RC) remains high, treatment strategies should be tailored to individual subtype and patient characteristics. Surgery plays a more significant role in clear cell renal cell carcinoma (ccRCC), particularly when detected early, as early detection allows for more effective surgical intervention. Minimally invasive treatments, such as thermal ablation or active surveillance, may be more suitable for subtypes with good prognoses.”

Table1: some data are obscured. Suggest to provide a table with clearer data. Suggest to convert table to pdf version before publication. Done

Mortality analysis: It would be good to add multivariate analysis that could calculate the cancer specific mortality risk for someone with ccRCC compared to those diagnosed with non-ccRCC We appreciate the reviewer’s suggestion to include a multivariate analysis calculating the cancer-specific mortality risk for patients with clear cell renal cell carcinoma (ccRCC) compared to those with non-clear cell renal cell carcinoma (non-ccRCC). Results are reported in Table 2 of the manuscript.

Overall survival

Line 220: non-ccRCC demonstrated better OS than ccRCC (5-year survival (5-ySR) 72.6%, 95% 70.3-74.7%

- the result writing for non-ccRCC should be looked at again We apologize for the unclear presentation of the overall survival data in the previous version of the manuscript. To improve clarity, we have revised the text as follows (page 12, lines 439-446)

Specifically: “Overall survival (OS) differed significantly between histological subtypes. In univariate analysis, non-ccRCC demonstrated better OS (5-year survival (5-ySR) 72.6%, 95% CI 70.3-74.7%) than ccRCC (5-ySR 62.7%, 95%CI 61.8-63.5) (Fig 2a). However, the OS differs significantly among the different non-ccRCC subtypes. The 5-ySR for pRCC is 74.3% ( 95%CI 71.2-77.2), and for chRCC 82.2% (95% CI 77.2-86.1) is better compared to ccRCC. Other known variants appear to have similar OS to ccRCC 5-ySR 72.9 % (95%CI 67.9-77.2). In contrast sarcRC and CDC had significantly lower OS, with 5-ySR 29.2% (95% CI 20.6-38.3) and 23.5% (95% CI 7.3-44.9). (Fig 2b.)”

Table3: the table provided is still raw. Suggest to polish the table for publication. The surgical treatment data could not account for the various indications for surgery or if the tumour was inoperable, so this should be mentioned as a limitation. Also, it would be good to compare the survival outcomes for patients receiving total vs partial nephrectomy if these data are available We appreciate the note on table 3, we decided to omit the table. We made the following change to the manuscript on page 12, lines 584-586:” Although a trend towards improved survival with surgery was observed in non-ccRCC subtypes combined (aHR1.29, 95% CI 0.94-1.76) and as separate entities, these associations did not reach statistical significance.”

In order to address the valid point on partial and total nephrectomy. We have added a new table 3 of multivariable adjusted Cox regression models to the manuscript on lines 594-596, page 14.

The new table 3 is in this response under the name table 2, due to the order it is addressed.

We added the following changes to clarify the addition made in the manuscript on page 14, lines 587-590:” Comparison of conservative treatment with total and partial nephrectomy demonstrated that patients undergoing partial nephrectomy achieved superior survival outcomes. Moreover, the findings reaffirmed that surgical intervention provided the greatest benefit for patients with ccRCC (Table 3).”

We appreciate the reminder on possible limitations in treatment allocations, and we added it to the manuscript on page 15 on lines 597-598 :” As this study is retrospective, we can expect a clinician bias in allocating patients for their preferred management.”

Discussions: Largely adequate.

Explanation that support stage downshift will improve the discussion further. Add more references that cover this aspect to strengthen the discussion. We appreciate the notion, to further expand this area for discussion. We have added references and made the following change on page 16, lines 704-721: “Stage migration and its impact

We observed improved RCC survival over time, with patients diagnosed after 2000 having better prognoses than those diagnosed in the 1990s. This likely reflects stage migration towards a more localized RC due to the increased usage of imaging and more efficient treatment modalities. A study in East of England observed a 16 % increase in stage I from 1999-2003 vs 2014-2016), with a 11% decrease in the same periods in stage IV (or missing stage.)[12]

This phenomenon, known as stage downshift, has been widely reported in the literature.[11,12,30-34]

Importantly, research has shown a differential impact of stage shift on clear cell renal cell carcinoma (ccRCC) and non-clear cell renal cell carcinoma (non-ccRCC).[32-34] Also, our data support that stage downshift is more pronounced for cc

---

## [Decision Letter · Decision Letter 1]

Renal cancer survival in clear cell renal cancer compared to other types of tumor histology: a population-based cohort study

PONE-D-24-58657R1

Dear Dr. Sepp,

We’re pleased to inform you that your manuscript has been judged scientifically suitable for publication and will be formally accepted for publication once it meets all outstanding technical requirements.

Kind regards,

Yuki Arita, M.D., Ph.D

Academic Editor

PLOS ONE

Additional Editor Comments (optional):

Reviewers' comments:

Reviewer's Responses to Questions

**Comments to the Author**

Reviewer #1: All comments have been addressed

Reviewer #2: All comments have been addressed

2. Is the manuscript technically sound, and do the data support the conclusions?

Reviewer #1: Yes

Reviewer #2: Yes

3. Has the statistical analysis been performed appropriately and rigorously?

Reviewer #1: Yes

Reviewer #2: Yes

4. Have the authors made all data underlying the findings in their manuscript fully available?

Reviewer #1: Yes

Reviewer #2: Yes

5. Is the manuscript presented in an intelligible fashion and written in standard English?

Reviewer #1: Yes

Reviewer #2: Yes

Reviewer #1: (No Response)

Reviewer #2: Abstract, Introduction, Methodology: All well-written & adequate. Suggest to observe choice of font because different fonts were used at different areas of the methodology

Results: Improved, well-written & adequate.

Discussion: Largely adequate and well-written. Suggest to observe choice of font because different fonts were used at different areas of the discussion.

Obvious improvements:

- The subheadings helped to compartmentalise the discussion.

- The clinical interpretation of the results were discussed in more detailed.

- Reviewer comments were incorporated into the write-up.

- Authors also more transparent about missing data & biases.

- Practical implications were clearly outlined.

References: Additional ~10 references were added.

**Do you want your identity to be public for this peer review?** For information about this choice, including consent withdrawal, please see our Privacy Policy

Reviewer #1: No

Reviewer #2: **Yes: ** INTAN SUHANA ZULKAFLI

---

## [Editor Report · Acceptance letter]

PONE-D-24-58657R1

PLOS ONE

Dear Dr. Sepp,

I'm pleased to inform you that your manuscript has been deemed suitable for publication in PLOS ONE. Congratulations! Your manuscript is now being handed over to our production team.

Kind regards,

on behalf of

Dr. Yuki Arita

Academic Editor

PLOS ONE